# Brain organoid formation on decellularized porcine brain ECM hydrogels

**Robin Simsa**[1,2,3‡]*, **Theresa Rothenbücher**[4‡], **Hakan Gürbüz**[5,6], **Nidal Ghosheh**[7], **Jenny Emneus**[5], **Lachmi Jenndahl**[1], **David L. Kaplan**[3], **Niklas Bergh**[2]*, **Alberto Martinez Serrano**[4‡], **Per Fogelstrand**[2‡]

**1** VERIGRAFT AB, Gothenburg, Sweden, **2** Department of Molecular and Clinical Medicine/Wallenberg Laboratory, University of Gothenburg and Sahlgrenska University Hospital, Gothenburg, Sweden, **3** Department of Biomedical Engineering, Tufts University, Medford, MA, United States of America, **4** Department of Molecular Biology, Center of Molecular Biology "Severo Ochoa" (UAM-CSIC), Universidad Autonoma de Madrid, Madrid, Spain, **5** Department of Biotechnology and Biomedicine, Technical University of Denmark, Kgs. Lyngby, Denmark, **6** FELIXROBOTICS BV, Utrecht, Netherlands, **7** School of Bioscience, Systems Biology Research Center, University of Skövde, Skövde, Sweden

‡ Shared first authorship. Shared senior authorship.
* robin.simsa@verigraft.com (RS); niklas.bergh@gu.se (NB)

**Data Availability Statement:** All relevant data are within the manuscript and its Supporting Information files.

**Funding:** This study was supported by the European Union's Horizon 2020 Research and

## Abstract

Human brain tissue models such as cerebral organoids are essential tools for developmental and biomedical research. Current methods to generate cerebral organoids often utilize Matrigel as an external scaffold to provide structure and biologically relevant signals. Matrigel however is a nonspecific hydrogel of mouse tumor origin and does not represent the complexity of the brain protein environment. In this study, we investigated the application of a decellularized adult porcine brain extracellular matrix (B-ECM) which could be processed into a hydrogel (B-ECM hydrogel) to be used as a scaffold for human embryonic stem cell (hESC)-derived brain organoids. We decellularized pig brains with a novel detergent- and enzyme-based method and analyzed the biomaterial properties, including protein composition and content, DNA content, mechanical characteristics, surface structure, and antigen presence. Then, we compared the growth of human brain organoid models with the B-ECM hydrogel or Matrigel controls *in vitro*. We found that the native brain source material was successfully decellularized with little remaining DNA content, while Mass Spectrometry (MS) showed the loss of several brain-specific proteins, while mainly different collagen types remained in the B-ECM. Rheological results revealed stable hydrogel formation, starting from B-ECM hydrogel concentrations of 5 mg/mL. hESCs cultured in B-ECM hydrogels showed gene expression and differentiation outcomes similar to those grown in Matrigel. These results indicate that B-ECM hydrogels can be used as an alternative scaffold for human cerebral organoid formation, and may be further optimized for improved organoid growth by further improving protein retention other than collagen after decellularization.

Innovation Program under the Marie Sklodowska-Curie Grant (No. 722779) awarded to RS; AMS group (CBMSO-UAM) grant (No. MINECO SAF-2017-83241-R) awarded tp AMS, ISC-III RETICS TerCel (RD16/0011/0032) and ASCTN597 Training (No. 813851) awarded to AMS, and NIH grants ((P41EB027062 and R01NS092847) awarded to DLK. VERIGRAFT AB provided funding in the form of salary for RS. Felix Printers provided funding in the form of salary for HG. The specific roles of these authors are articulated in the 'author contributions' section. The funders had no role in study design, data collection and analysis, decision to publish, or preparation of the manuscript.

# Introduction

The development of brain organoids for studies on neurodevelopment and disease has emerged as a novel research field [1]. For example, brain organoids have been used for experimental studies on microcephaly [2], schizophrenia [3], zika virus [4], and Alzheimer's disease [5]. Brain organoids are three-dimensional, neuronal tissues generated from human pluripotent stem cells after embryoid body formation [2]. The differentiation and maturation of neuronal tissue within brain organoids is induced by adding differentiation factors to the culture medium. During this process, aggregated cells organize into a 3D spherical structure, with a partly brain-like architecture with ventricular zones as well as deep- and upper-layer neurons [2]. Besides these cortical structures, more specific brain regions can be developed in the organoids, such as the midbrain and hypothalamus [4, 6, 7].

Most commonly, Matrigel is used as an outer hydrogel scaffold to give brain organoids structural support and to supply the organoid tissue with differentiation factors [8]. However, the composition of Matrigel differs from the composition of the extracellular matrix (ECM) of the native brain and is therefore not specific [9]. Potential alternatives to Matrigel are ECM hydrogels, which are obtained by decellularization of the native brain.

The ECM consists of structural and functional proteins that give cells structural support and guide cell behavior. The composition of the ECM is specific for each tissue type and contributes to the individual functions of every tissue [10]. The ECM of the brain is unique in its composition, as it contains less fibrous structural proteins, such as collagen or elastin, and more proteoglycans compared to other tissues [11, 12]. In addition to ECM, some proteoglycans are also anchored in cell membranes, where they regulate growth factor availability to cells, e.g. by binding of nerve growth factor (NGF), vascular endothelial growth factor (VEGF), or platelet-derived growth factor (PDGF) to proteoglycan carbohydrate side-chains, the glycosaminoglycans (GAGs) [13]. Proteoglycans profoundly influence neuronal cell behavior by regulating cell adhesion and neurite outgrowth [11]. Lecticans, which are a subfamily of proteoglycans, are especially abundant in the brain ECM [11, 12]. Four members of the lectican family have been identified in different splicing variants in the brain, namely brevican and neurocan, which are unique for brain ECM, and aggrecan and versican, which exist in a variety of tissues [11, 14]. The ECM composition is furthermore unique for specific brain regions, suggesting that the brain ECM influences neuronal cell function. The brain ECM is continuously regulated *in vivo* by cell-secreted enzymes, such as matrix metalloproteinases (MMPs), which contribute to plasticity and dynamic remodeling of the brain. The total contribution of the ECM in the brain in terms of mass or volume is low compared to other tissues, reported to be ~20% of the total weight of the brain [14, 15]. Due to the unique composition of brain ECM, ECM hydrogels from decellularized brain tissue may be a more specific scaffold than those derived from Matrigel for *in vitro* differentiation of embryonic stem cells and human neural tissues.

Decellularization (DC) is the process of removing cells from the ECM by chemical, physical, and enzymatical methods [16]. DC generates cell-free ECM scaffolds, with minimal interference from cellular antigens and metabolites. However, the DC process exhibits a risk of damaging ECM proteins, and thus requires a balance between sufficient cell removal and preserved integrity of the ECM. DC has been applied on multiple tissue types, such as blood vessels [17, 18], heart [19], lung [20], kidney [21], and also brain [22–30]. Once decellularized, the ECM can be processed into hydrogels with a variety of methods [31]. The basic principle of hydrogel formation is the solubilization of ECM proteins with acids and enzymes. Commonly, pepsin is used for that purpose, which cleaves the nonhelical protein regions outside of the triple helix protein structure of collagen (= telopeptides) that form the intramolecular bonds between

collagen fibrils, therefore producing monomeric components [32]. After neutralization to physiological pH, pepsin gets inactivated and the hydrogel formation then follows a collagen-based self-assembly process at 37°C or below [31]. B-ECM hydrogels have previously been reported to enhance neuronal maturation compared to conventional substrates such as collagen or laminin [22, 30] and have been used for site-specific differentiation effect on neuronal stem cells and induced pluripotent stem cells [33]. However, B-ECM hydrogels have not yet been used as a scaffold for cerebral organoid formation.

In the present study, we evaluated the use of decellularized whole pig brain ECM hydrogels as a scaffold for cerebral organoid growth, in comparison to the commonly applied Matrigel.

## Materials and methods

### Decellularization of porcine brain

Frozen whole pig brains (FarmLand, USA) were obtained via Amazon.com and stored at -20°C. Brains were derived from adult animals (6 months at the time of slaughter), mixed gender, of Smithfield breed, raised in the USA. Brains were delivered in a sealed plastic cup without any additives or preservatives. For decellularization, the brain tissue was thawed, cut into small pieces with a scalpel, and added to a 1L glass bottle. The total tissue volume for one batch was about 400g per bottle. The tissue pieces were then washed with $H_2O$ for 4–5 h in an incubator at 37°C under agitation at 150 rpm (New Brunswick Scientific Model G25), which resulted in a thick slurry. For changing of reagents, the liquid was removed by decanting over a porcelain filter (Porcelain Bitumen Filter Crucible, Cole-Parmer, USA) to retain brain tissue, with slight stirring with a metal spatula. The brain tissue was then incubated with 1% sodium deoxycholate (SDC, Sigma, USA). The SDC solution was changed twice daily. After 2–3 days, the liquid usually was transparent and the remaining brain tissue volume remained constant, indicating washout of most blood and other soluble components. After 4 days of SDC treatment, the tissue was washed for 4–5 h in $H_2O$ and added to 125 mL 40 U/mL deoxyribonuclease I (DNase, Sigma, USA) for incubation o/n. Finally, the tissue was washed in $H_2O$ for 4-5h with several liquid changes, and the remaining tissue was collected. The final tissue weight was measured, biopsies for analysis were collected, and the remaining tissue (B-ECM) was frozen in a 50 mL falcon at -80°C and then lyophilized.

### B-ECM solubilization and hydrogel formation

Lyophilized brain tissue was solubilized following the method of Freytes et al. [34]. Briefly, B-ECM was powdered with a mill (Wiley Mini Mill, 3383-L10, Thomas Scientific, USA) over a 60 mesh (equal to 250 μm particle size). Then, a desired volume of 0.1 M HCl solution diluted in $H_2O$ was prepared, and pepsin (Sigma, USA) and B-ECM powder added at a ratio of 1:10 (e.g. 1 mg/mL Pepsin together with 10 mg/mL B-ECM). After 72h incubation at room temperature with a magnetic stirrer, the solution was centrifuged at 13,000 rpm for 10 min, and the supernatant consisting of the solubilized B-ECM was collected. The solubilized B-ECM was stored at 4°C until the solution was cooled down. The B-ECM solution was then kept on ice and cold 10x PBS was added to make up a volume of $1/10^{th}$ of the total volume, resulting in a concentration of 1x PBS. While still on ice, the pH of the solution was adjusted to 7.2–7.8 with cold NaOH, and the solution was quickly frozen and later lyophilized. Then, the lyophilized tissue was powdered once more over a 60 mesh and stored until use. Directly before use, the neutralized B-ECM powder was dissolved in cold $H_2O$ at the desired concentration and strongly vortexed, which led to hydrogel formation at 37°C. In this study, analysis on decellularized B-ECM are labeled as "B-ECM", and analysis on fully formed B-ECM hydrogels are labeled as "B-ECM hydrogel".

## DNA quantification

DNA from 10–30 mg wet tissue was extracted using the DNeasy Blood and Tissue Kit (Qiagen, Germany), following the manufacturer's protocols. Briefly, native brain tissue or B-ECM samples (n = 9) were incubated at 55˚C with proteinase K solution until the tissue was completely digested. DNA was extracted with spin columns and quantified with a Qubit 3.0 Fluorometer (ThermoFisher, USA). The total amount of DNA was calculated from the supplied DNA standard.

## Histology and immunohistochemistry

Samples of native brain, B-ECM or B-ECM hydrogel were fixated in 4% paraformaldehyde for 1h for fixation, embedded in paraffin, and cut in 5 μm sections (n = 3). The sections were deparaffinized at 60˚C for 1h and rehydrated following standard protocols before use.

Nuclei in the native brain and B-ECM samples (sections and whole surface) were stained with 25 μg/mL DAPI for 5 min. Histological staining with hematoxylin-eosin (H&E) was performed following standard procedures. Pictures for DAPI or H&E stained samples were taken with a fluorescent microscope (Keyence, Japan). For immunohistochemical staining, antigen retrieval was performed in a Tris-EDTA buffer in a water bath at 95˚C for 15 min. Samples were then incubated for 30 min at room temperature with FBS for blocking. For IHC, the following antibodies were used: anti-fibronectin (1:400, #ab23750, Abcam, UK), anti-vitronectin (1:200, #ab13413, Abcam, UK), anti-myelin basic protein (MBP, 1:100, #ab124493, Abcam, UK), laminin (1:100, #ab11575, Abcam, UK), and anti-human leucocyte antigen (HLA-DR, 1:100, #sc53302, Santa Cruz, USA). All antibodies were diluted in PBS. Following overnight incubation at 4˚C in a wet chamber, slides were washed in PBS and incubated with secondary antibody conjugated to AlexaFluor488 (donkey-anti mouse, #A-21202, ThermoFisher) or conjugated to AlexaFluor594 (donkey-anti rabbit, #A-21203, ThermoFisher) at a concentration of 1:200. Finally, samples were mounted with Prolong antifade (Fisher Scientific) and imaged with a fluorescence microscope (Apotome Axioplan 2, Zeiss).

## Scanning electron microscopy

Scanning electron microscopy (SEM) was performed on samples of native brain, B-ECM or B-ECM hydrogel. The samples were dehydrated in a series of 35%, 60%, 80%, 90%, 95%, and 100% ethanol in time intervals of 10 minutes each, followed by incubation with hexamethyldisilazane (Sigma, USA) for 10 min. The liquid was aspirated, and the samples were air-dried. Prior to SEM imaging, samples were sputter-coated with gold for 120 sec and imaged under vacuum with a scanning electron microscope (Zeiss EVO MA10, Germany).

## Quantification of collagen and glycosaminoglycans

Collagen and glycosaminoglycans (GAGs) in the native brain and B-ECM samples were quantified with commercial kits. For the quantification of collagen, which can be solubilized with the hydrogel preparation technique (described above), the Sircol soluble collagen assay (Biocolor, United Kingdom) was used. Briefly, 10–20 mg samples (n = 15) were dissolved in 1 mL 0.1 M HCl containing 1 mg pepsin o/n at room temperature. Then, samples were incubated in the supplied collagen isolation and concentration reagent o/n at 4˚C, after which the supplied dye reagent was added. The absorbance of 200 μL sample was measured spectroscopically at 555 nm. The concentration was determined from a standard of known concentration. For GAG quantification, the glycosaminoglycan assay (Chondrex, USA) was used, which detects sulfated GAGs. 15–30 mg samples (n = 8) were dissolved in a 0.2 M sodium phosphate buffer

containing 5 mM cysteine HCL, 5 mM EDTA and 125 μg/mL Papain o/n at 65˚C. Then, samples were centrifuged at 13,000 rpm (Eppendorf Centrifuge 5417R) and the supplied dye reagent added to the supernatant. Absorbance was measured spectroscopically at 525 nm. Following the individual supplier's instructions, collagen was measured from dry sample weight and GAGs from wet sample weight.

## Mass spectrometry

Mass Spectrometry (MS) of native brain, B-ECM, B-ECM hydrogel and Matrigel was performed by standard shotgun proteomics analysis. For brain samples, 6 individual batches were combined to a pooled individual sample. Lysate preparation and digestion was done according to previously published methods [35]. Briefly, samples were lysed using 20 μL of lysis buffer (consisting of 6 M guanidinium hydrochloride, 10 mM Tris(2-carboxyethyl)phosphine (TCEP), 40 mM carbamoylamino acids (CAA), 50 mM HEPES pH8.5). Samples were boiled at 95˚C for 5 minutes, after which they were sonicated for 3x 10 sec in a Bioruptor sonication water bath (Diagenode) at 4˚C. After determining protein concentration with a Bradford assay (Sigma), 50 μg was taken forward for digestion. Samples were diluted 1:3 with 10% acetonitrile, 50 mM HEPES pH 8.5, then LysC (MS grade, Wako) was added in a 1:50 (enzyme to protein) ratio, and samples were incubated at 37˚C for 4 h. Samples were further diluted to 1:10 with 10% Acetonitrile, 50 mM HEPES pH 8.5. Trypsin (MS grade, Promega) was added in a 1:100 (enzyme to protein) ratio and samples were incubated overnight at 37˚C. Enzyme activity was quenched by adding 2% trifluoroacetic acid (TFA) to a final concentration of 1%. Prior to mass spectrometry analysis, the peptides were desalted on in-house packed C18 StageTips [36]. For each sample, 2 discs of C18 material (3M Empore) were packed in a 200 μL tip, and the C18 material activated with 40 μl of 100% Methanol (HPLC grade, Sigma), then 40 μl of 80% acetonitrile with 0.1% formic acid. The tips were subsequently equilibrated 2x with 40 μL of 1% TFA, 3% Acetonitrile, after which the samples were loaded using centrifugation at 4.000x rpm. After washing the tips twice with 100 μL of 0.1% formic acid, the peptides were eluted into clean 500 μL Eppendorf tubes using 40% acetonitrile with 0.1% formic acid. The eluted peptides were concentrated in an Eppendorf Speedvac, and reconstituted in 1% trifluoroacetic acid, 2% acetonitrile for Mass Spectrometry (MS) analysis.

MS data acquisition was performed by loading peptides of each sample onto a 2 cm C18 trap column (ThermoFisher 164705), connected in-line to a 50cm C18 reverse-phase analytical column (Thermo EasySpray ES803) using 100% Buffer A (0.1% Formic acid in water) at 750 bar, using the Thermo EasyLC 1000 HPLC system, and the column oven operating at 45˚C. Peptides were eluted over a 260 minute gradient ranging from 6 to 60% of 80% acetonitrile, 0.1% formic acid at 250 nL/min, and the Q-Exactive instrument (Thermo Fisher Scientific) was run in a DD-MS2 top10 method. Full MS spectra were collected at a resolution of 70,000, with an AGC target of 3×106 or maximum injection time of 20 ms and a scan range of 300–1750 m/z. The MS2 spectra were obtained at a resolution of 17,500, with an AGC target value of $1 \times 10^6$ or maximum injection time of 60 ms, a normalised collision energy of 25 and an intensity threshold of $1.7^4$. Dynamic exclusion was set to 60 s, and ions with a charge state <2 or unknown were excluded. MS performance was verified for consistency by running complex cell lysate quality control standards, and chromatography was monitored to check for reproducibility.

The raw files were analysed using Proteome Discoverer 2.4. Label-free quantitation (LFQ) was enabled in the processing and consensus steps, and spectra were matched against the H*uman Matrisome* and the *Mouse Matrisome* database obtained from the matrisome project (matrisomeproject.mit.edu). Dynamic modifications were set as Oxidation (M), Deamidation

(N,Q) and Acetyl on protein N-termini. Cysteine carbamidomethyl was set as a static modification. All results were filtered to a 1% false discovery rate (FDR), and protein quantitation done using the built-in Minora Feature Detector.

The results of detected proteins from the different samples (S1 File) were merged by the uniprot accession ID for the samples of native, decellularized and hydrogel pig brain. The gene names for the Uniprot accession IDs were retrieved applying the *Retrieve/ID mapping* tool on the Uniprot homepage (www.uniprot.org). Thereafter, the data set was merged with the results of detected proteins from the Matrigel sample, ordered by gene name. The normalized abundances for each sample was log 2 transformed and the resulting data set was quantile normalized applying the *normalize.quantile()* function in package *preprocessCore* in R. The normalized protein abundances were then filtered to include only proteins that were identified with at least 2 unique peptides (S2 File). To explore the ECM content of the data set, the filtered data set was subsequently merged with the H*uman Matrisome* (samples native, decellularized and hydrogel) or the *Mouse Matrisome* (matrigel) of the Matrisome Project (matrisomeproject.mit.edu). To identify the overlapping Matrisome proteins from the different samples, Venn diagram was constructed applying the function *venn.diagram()* in package *VennDiagram* in R (S2 Fig). To extract the proteins from the different areas of the Venn diagram the function *calculate.overlap()* was applied (S3 File).

## Rheology

To obtain stiffness characteristics of the B-ECM hydrogel, rheological measurements were performed with an ARES-LS2 Shear Rheometer (TA Instruments, USA). For all experiments, neutralized B-ECM solutions at different concentrations were stored on ice to prevent premature gelling before assaying. The 25 mm parallel plates were cooled to 4˚C with a Peltier unit, then B-ECM solution was added. A 1 mL sample (n = 3) was added, and the gap distance between the plates set to 1mm. Mineral oil was applied to the borders of the liquid to prevent evaporation. Dynamic time sweep test was performed at 37˚C and settings of 1 rad/s and 5% strain. Following the dynamic time sweep test, a dynamic frequency sweep test was performed at a constant strain of 1% along a frequency range of 1 rad/s to 200 rad/s. After the dynamic frequency sweep test, a dynamic strain sweep test was performed at a constant frequency of 1 rad/s along a strain range of 0.1% to 200%.

## Turbidimetrics

Turbidimetric kinetic analysis was preformed similar to previously described methods [37, 38] to determine gelation rates. Neutralized B-ECM solution was prepared at different concentrations and kept on ice at 4˚C until the measurement was started. The spectrophotometer was preheated at 37˚C and 100 μL of B-ECM solution at different concentration (n = 6 replicates) was added to the wells. The absorbance at 405 nm was measured at 1 min intervals for 90 min and normalized between 0 ($A_0$; the initial absorbance) and 1 ($A_{max}$; the maximum absorbance).

$$\text{Normalized Absorbance (NA)} = \frac{A - A_0}{A_{max} - A_0}$$

Furthermore, the time when the absorbance reached 50% or 95% of respective maximum absorbance ($t_{1/2}$ or $t_{95}$) was noted. The gelation rate (S) was determined as the slope of the linear region of the gelation curve, and the lag time ($t_{lag}$) was defined as the intercept of the linear region of the gelation curve with 0% absorbance.

## Cell culture

Cell studies conducted in the present work were approved by the UAM Research Ethics Committee (Refs. CEI-74-1338 and CEI-96-1768), and then authorized by the Madrid Local Government Health Council under references 47/312421.9/17 and 07/654525.9/19. Human embryonic stem cell (hESC) line H9 (WA09, WiCell Research Institute Inc.) was cultured under feeder- and xenogen-free conditions on human recombinant laminin LN521 (BioLamina, #LN521-02) with Nutristem hPSC XF medium (Biological Industries, #05-100-1A). Cells below passage 60 were used for this study and regularly tested for mycoplasma and karyotype abnormalities and found negative. When reaching around 60% confluence, hESCs were collected for generating brain organoids.

## Methodology for brain organoid generation

For brain organoid formation, the STEMCELL Technologies "STEMdiff Cerebral Organoid Kit" (#08570) was used to guarantee experimental standardization. On day 0, feeder and xenogen-free cultured hESCs (H9) were detached when reaching 60–80% confluence and seeded with the supplied EB-formation medium as single cells (10.000 cells per well) into 96 v-bottom well plates (Falcon, #353263) to generate embryoid bodies. When the EBs reached day 5, the medium was changed to the supplied neural induction medium. After neural induction, brain organoids were divided into two groups. For including EBs in the hydrogel matrices, EBs were placed in prepared parafilm wells [39] and 15μL of pre-hydrogel solution was added in one drop on top of each EB. EBs of the first group were included in hESC-qualified Matrigel (Corning Matrigel hESC-qualified matrix, #354277) and the second group in B-ECM hydrogel (15 mg/mL). For both groups, polymerization was performed at 37˚C for 30 min. Brain organoids were cultured in the supplied expansion medium for 3 days, then media was switched to the supplied maturation medium and culture conditions were changed from static to agitated on an orbital shaker (throw: 10mm, speed: 105 rpm) until they reached 40 days of culture.

## qPCR

To test the effect of B-ECM hydrogel on brain organoids, brain organoids were cultured in the presence of B-ECM hydrogel (n = 13) or Matrigel (n = 12) in technical triplicates. After 40 days of cultivation, RNA was extracted from organoids using RNeasy Mini Kit (Qiagen, #74104). RNA contents were quantified by using NanoDrop One. Reverse transcription (RT) reactions were performed using the iScript cDNA Synthesis kit (Biorad PN170-8891) following manufacturer´s instructions. Briefly, 1000 ng of total RNA from each sample were combined with 5 μl of master mix (includes all necessary reagents among which a mixture of random primers and oligo dT for priming). The reaction volume was completed up to 20 μl with DNAse/RNAse free distilled water (50 ng/μl) (Gibco PN 10977). Thermal conditions consisted of the following steps: 5 min 25˚C, 20 min 46˚C and 1 min 95˚C. A melting curve from 60˚C to 95˚C (0.5˚C/sec) was included additionally at the end of the program to verify the specificity of the PCR. Fluorescence was acquired during both the 60˚C and melting steps. The genes of interest (GOI) analyzed were NESTIN, TUBB3, DCX, MAP2 and GFAP, with specific primers (See S1 Table). Results were normalized with selected reference genes (GAPDH, ACTB, ATP1B1, 18S, TBP) and statistical analysis was by t-test.

## Immunohistochemistry of brain organoids

Forty day old brain organoids were fixed with buffered 4% paraformaldehyde (20min, RT), followed by 3 washing steps in PBS. As preparation for cryosectioning, organoids were

equilibrated in 30% sucrose solution for 1h and subsequently embedded in Tissue Tek O.C.T (Sakura, Netherlands). For immunohistochemistry brain organoids were cut into 20 μm sections with a cryostat. The obtained sections were incubated at 60˚C for 15 min and pre-treated with 1% sodium dodecyl sulfate (SDS) solution for 5 min. Blocking was done for 1.5 h with 10% donkey serum in PBS buffer containing 0.5% Triton X-100. Primary antibodies were diluted in 1% donkey serum in 0.5% Triton X-100 containing PBS buffer. Primary antibodies for neuronal progenitors were anti- Nestin (1:100, #611658, BD Transduction, USA), PAX6 (1:100, #901301, Biolegend, USA), SOX2 (1:200, #AB5603, Millipore). For more mature neurons β-Tubulin III (1:500, #T2200, Sigma, USA), MAP-2 (1:300, #M4403, Sigma, USA), and DCX (1:300, #SC-8066, Santa Cruz, USA). Primary antibodies were incubated over night at 4˚C. Next day sections were washed two times with PBS buffer containing 0.5% Triton X-100. Secondary antibodies were added and incubated for 2 h at room temperature: Anti-rabbit Alexa 488 (1:500, Molecular probes, #A-11008, USA), anti-rabbit [Alexa 647] (1:500, Molecular probes, # A-31573, USA) anti-mouse Alexa 647 (1:500, Molecular Probes, # A-31571, USA), anti-goat FITC (1:500, Molecular Probes, #A16000 USA]; DAPI (Santa Cruz Biotechnology, #28718-90-3) was used as nuclear counterstaining. Sections were mounted with MOWIOL (inhouse) and imaged with confocal microscopy (LSM710, Zeiss).

### Statistical analysis

Statistical analysis was performed with GraphPad Prism 8. For comparisons of 2 groups, unpaired t-test was used. For comparison of multiple groups, one-way ANOVA and Tukey´s multiple comparison tests were applied. Significance is indicated in bar graphs with asterisk signs (*), indicating p-value of $p \leq 0.05$ (*), $p \leq 0.01$ (**) or $p \leq 0.001$ (***). No significance (n.s.) is displayed for $p > 0.05$.

## Results

### Pig brain decellularization and hydrogel formation

Native pig brains were obtained from a commercial supplier, delivered frozen in plastic cups, hence the cylindrical form (Fig 1A). The brains were cut into small pieces and decellularized with a novel SDC-, and DNase based protocol. Following DC, hydrogel formation at different concentrations was achieved by acid-pepsin digestion of ECM proteins and subsequent neutralization. DC with this protocol led to an approximate decrease of 98.7% of the total native brain mass, resulting in remaining ECM wet weight of 6.21 ± 1.59 g per 450 g starting material. Histological staining with DAPI showed only a few or no nuclei in the DC samples (Fig 1B). Quantification of remaining DNA showed a significantly lower DNA content in the B-ECM (13.07 ± 7.86 ng/mg wet tissue) compared to the native ECM (142.3 ± 67.01 ng/mg wet tissue), which however indicates that some remnant DNA molecules are still present in the ECM (Fig 1C). The content of sulfated GAGs was significantly decreased in the DC sample (0.70 ± 0.33 μg/mg wet tissue) compared to the native sample (1.71 ± 0.63 μg/mg wet tissue) (Fig 1D). Due to cell removal, the soluble collagen content was significantly increased in the decellularized sample (113.5 ± 24.65 μg/mg dry weight) compared to the native sample (14.11 ± 1.15 μg/mg dry weight) (Fig 1E). Given these results, a hydrogel prepared at a concentration of 15 mg/mL with lyophilized B-ECM would contain a total amount of 1.70 mg/mL collagen. Taken together, these results show that the DC process removed more than 90% of the nuclear material and structural proteins such as collagen were retained. However, functional proteins such as GAGs were partly washed out during DC.

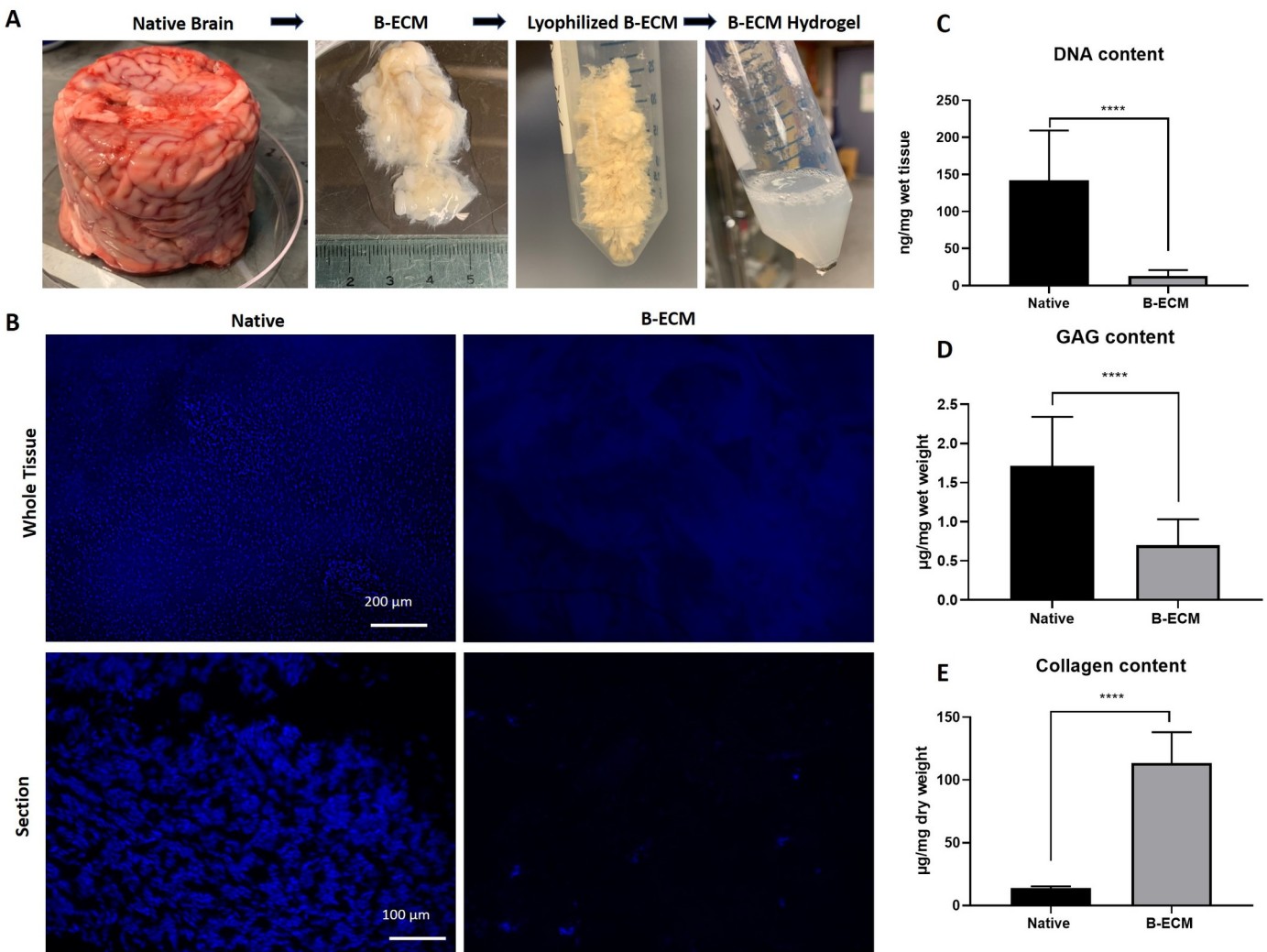

**Fig 1. DNA and ECM content of decellularization pig brain.** (A) Native pig brains were decellularized with detergents, lyophilized, and solubilized in a pepsin-HCl mix. After neutralizing the solution (15 mg/mL) to physiological salt and pH concentration, hydrogel formation at 37˚C was observed. (B) DAPI staining of whole tissue samples *en face* (upper panel) and tissue sections (lower panel). (C) DNA content in native brain and B-ECM samples (n = 9) was measured after extraction of DNA from wet tissue with a commercial kit. (D) GAGs from wet tissue (n = 8) and (E) collagen from dry tissue (n = 15) were quantified by extraction of proteins with commercial kits.

## ECM composition

ECM morphology was visualized with histology and immunohistochemistry (Fig 2). H&E staining confirmed the presence of cells in the native tissue, but did not show cells in the B-ECM or B-ECM hydrogel groups. We found that fibronectin was preserved in all groups, while less staining of vitronectin was evident in the B-ECM and B-ECM hydrogel groups. Laminin was preserved in the native and B-ECM samples but showed fragmentation in the B-ECM hydrogel. Both immunogenic cell surface markers MBP and HLA-DR were present in the native samples, but significantly decreased in the B-ECM and B-ECM hydrogel samples.

Mass spectrometry analysis showed that decellularization and hydrogel processing removed most proteins except for collagen, which made up more than 90% of total protein abundance in both the B-ECM and the B-ECM hydrogel group (Fig 3A and 3B, S1–S4 Files). In the native ECM, 101 proteins included in the matrisome (ECM proteins and ECM-associated proteins)

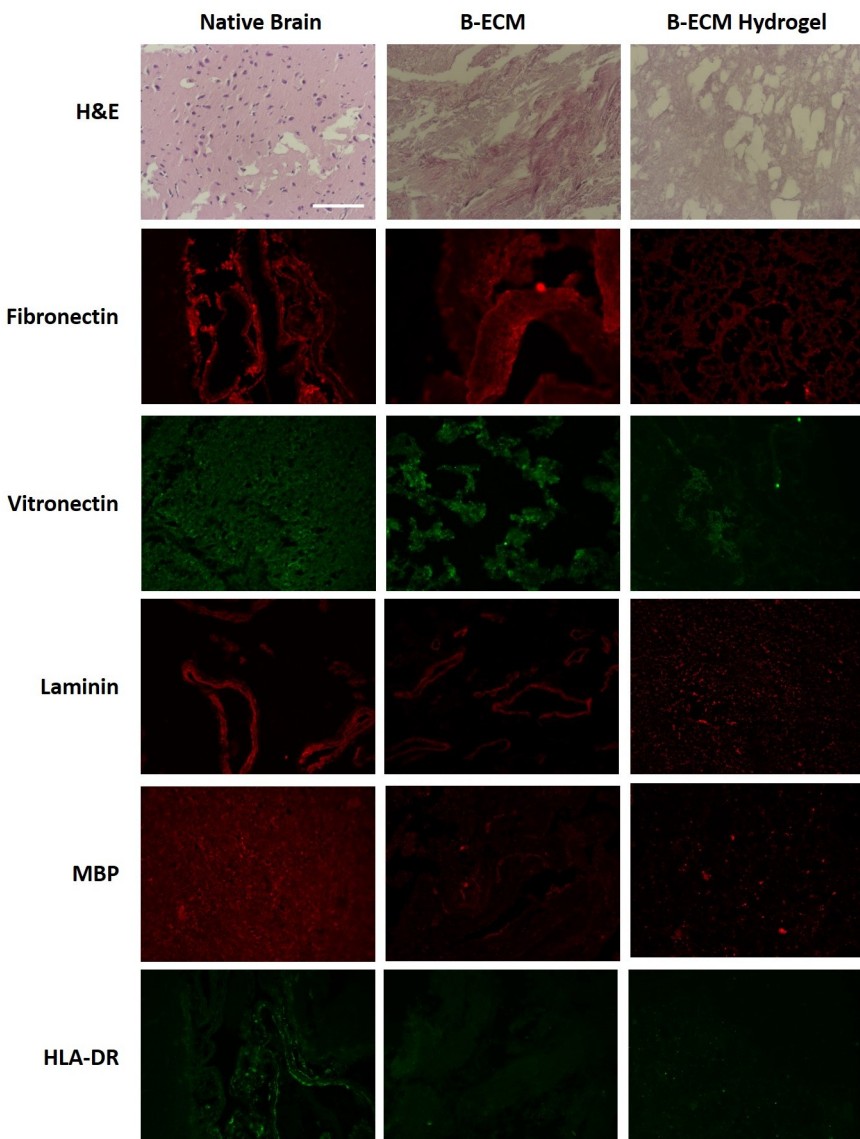

**Fig 2. Staining of native, B-ECM and B-ECM hydrogel samples.** Histological and immunostaining was performed on paraffin-embedded samples after rehydration and antigen retrieval (n = 3). For general morphology, the samples were stained with hematoxylin-eosin (H&E). Fibronectin, vitronectin, laminin, myelin basic protein (MBP) and human leucocyte antigen (HLA-DR) were detected with antibodies (immunofluorescence). Scale bar represents 100 μm.

were detected, of which 64 are ECM-associated proteins, and 37 are ECM proteins consisted mostly of glycoproteins and proteoglycans (S2 File). In B-ECM, 91 matrisome-proteins were detected, of which 60 are ECM proteins, including 16 collagens, 31 glycoproteins, and 13 proteoglycans (S3 File). The collagens account for about 90% of the matrisome-proteins abundance in B-ECM, while they account for less than 5% in the native ECM (Fig 3A and 3B). 50 matrisom- proteins overlapped between native ECM and B-ECM (S2 Fig Venn diagram). In B-ECM hydrogel, 18 matrisome-proteins were detected, of which 15 were ECM proteins including six collagens (COL1A1, COL1A2, COL2A1, COL3A1, COL5A2, COL6A1), that accounted for greater than 95% of the abundance of the matrisome-proteins, eight glycoproteins (FBN1, FBN2, FGB, LAMB1, LAMB2, LAMC1, SRPX, and VWF), and one proteoglycan

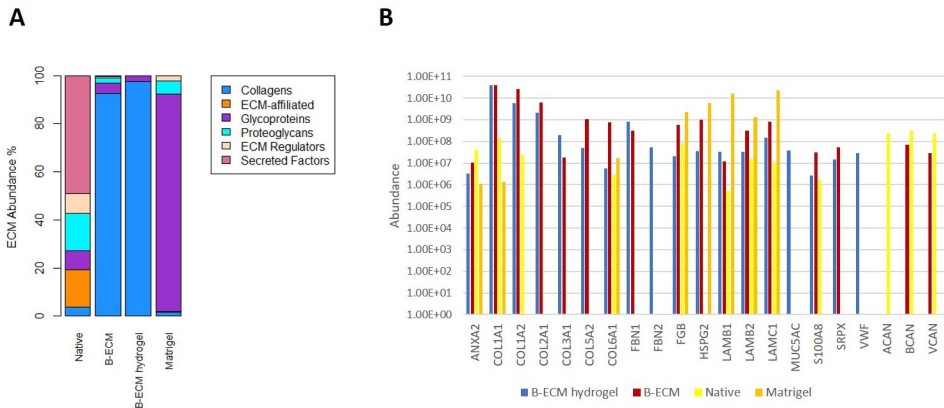

**Fig 3. Protein analysis performed by mass spectrometry on samples from native brain, B-ECM, B-ECM hydrogel, and on Matrigel.** (A) Stacked bar chart of the abundance of the matrisome-proteins that were detected in the analyzed samples, categorized in different matrisome categories (Collagens, ECM-affiliated, Glycoproteins, Proteoglycans, ECM Regulators or Secreted Factors) (B) The abundance of ECM proteins found in the B-ECM hydrogel sample compared to Native, B-ECM, and Matrigel samples, sorted by ECM categories (collagens, glycoproteins, and proteoglycans) with their gene identifier. Besides, the abundances of the brain-specific ECM proteins brevican (BCAN), aggrecan (ACAN), and versican (VCAN) in the different samples are shown as well. Y-axis is in the logarithmic scale.

(HSPG2) (Fig 3A and 3B). All ECM proteins except FBN2 and VWF overlapped between the B-ECM hydrogel and the B-ECM group (Fig 3B, S2 Fig, S3 File). Other brain-specific proteoglycans such as aggrecan (ACAN), brevican (BCAN), and versican (VCAN), were detected in the native samples, while only BCAN and VCAN were detected in B-ECM as well. None of these mentioned brain-specific proteoglycans were detected in the Matrigel or the B-ECM hydrogel. In the Matrigel, 75 matrisome -proteins were identified, of which 38 are ECM proteins, including ten collagens, 20 glycoproteins, and two proteoglycans (S2 File). The glycoproteins account for about 90% of the abundance of the matrisome proteins. 39 matrisome-proteins detected in Matrigel overlapped with the matrisome content of the different tissues from the pig brain. In total, 105 matrisome-proteins detected in the different tissues from the pig brain did not overlap with the matrisome content of the Matrigel (S2 Fig). In conclusion, mass spectrometry showed that most structural ECM proteins such as collagens were preserved, while several functional proteins were affected by decellularization and hydrogel processing. These results show a further need for optimization of decellularization and hydrogel processing strategies, in order to avoid the loss of functional proteins which may have beneficial effects on organoid development.

## Surface structure analysis

Surface structure analysis of native brain tissue, B-ECM and B-ECM hydrogel were analyzed with scanning electron microscopy (SEM) (Fig 4). While native brain tissue showed the presence of cells covering the surface, decellularized ECM was mostly free of cellular material and collagen fibers were clearly visible. The hydrogel also showed reorganization of collagen fibers in a more disordered, porous 3D structure. These results confirm the successful DC of brain tissue and the reorganization of brain ECM proteins following solubilization and neutralization.

## Rheological measurements and gelation kinetics

To obtain mechanical characteristics of the B-ECM hydrogels, rheological measurements were performed at different concentrations, similar to previous protocols [40, 41] and summarized

| Native | B-ECM | B-ECM Hydrogel |
|---|---|---|

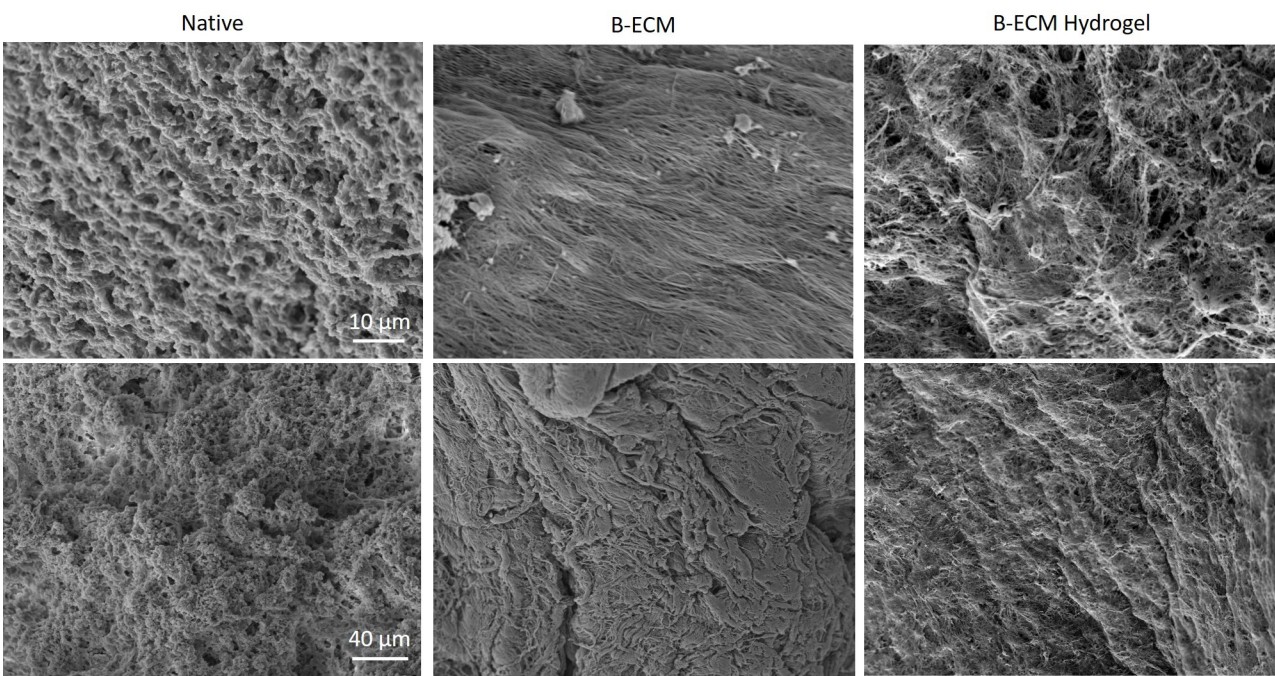

**Fig 4. SEM of native, B-ECM and B-ECM hydrogel samples.** Images taken with 4000x (upper panel) and 1000x (lower panel) magnification.

in Table 1. The hydrogels showed viscoelastic-solid behavior, as the storage modulus (G´, indicating elasticity) was higher than the loss modulus (G´´, indicating viscosity). The final stiffness of the B-ECM hydrogels was 0.7 Pa, 12.8 Pa, and 36.4 Pa for 5 mg/mL, 10 mg/mL and 15 mg/mL respectively (Table 1, S1 Fig). B-ECM showed an increased rate of gelation and higher storage modulus (more stiffness) with higher concentration. Dynamic strain sweeps were performed to determine the linear viscoelastic region of the hydrogels in relation to strain. The results showed that all three B-ECM hydrogel concentrations showed linear viscoelastic behavior up to 10% strain, as less than 5% change in elasticity (storage modulus) was detected within this range (S1B Fig). This value is similar to Matrigel, which has a reported linear viscoelastic behavior up to 13% strain [42].

The turbidimetric gelation kinetics of B-ECM hydrogels at concentrations of 10, 7.5 and 5 mg/mL were characterized spectrophotometrically and the gelation parameters at the

**Table 1. Analysis of mechanical properties and gelation kinetics of B-ECM hydrogels at different concentrations.**

| Rheology | | | |
|---|---|---|---|
| | **5 mg/mL** | **10 mg/mL** | **15 mg/mL** |
| Storage Modulus (G'; Pa) | 0.7 ± 0.05 | 12.8 ± 1.6 | 36.4 ± 6.5 |
| Loss Modulus (G"; Pa) | 0.09 ± 0.0008 | 2.1 ± 0.3 | 8.4 ± 0.6 |
| **Turbidimetric analysis** | | | |
| | **5 mg/mL** | **7.5 mg/mL** | **10 mg/mL** |
| Lag phase ($t_{lag}$; min) | 13.7 ± 3.7* | 8.5 ± 0.9* | 11.3 ± 1.2 |
| 50% gelation ($t_{1/2}$; min) | 23.5 ± 9.7 | 16.8 ± 1.1 | 15.2 ± 1.5 |
| 95% gelation ($t_{95}$; min) | 67.5 ± 11.6* | 59.2 ± 4.3 | 47.7 ± 2.8* |
| Speed (S; min) | 0.041 ± 0.017 | 0.044 ± 0.004 | 0.046 ± 0.002 |

Statistical differences in turbidimetric analysis between different rows are indicated with an asterix (*).

exponential phase of gelation were calculated (S1C Fig). The start of gelation phase, or lag phase ($t_{lag}$), was slower for the B-ECM hydrogels at 5 mg/mL ($t_{lag}$, 13.71 ± 3.69 min) than the hydrogels at 7.5 mg/mL ($t_{lag}$, 8.53 ± 0.95 min) and 10 mg/mL ($t_{lag}$, 11.27 ± 1.23 min). There was no statistically significant change in the time required to reach half the final turbidity ($t_{1/2}$) and the speed of the turbidimetric gelation kinetics (S), however a trend towards shorter times to reach gelation was observed. The maximum gelation time ($t_{95}$) took longer for lower B-ECM concentrations. These results determined the hydrogel assembly kinetics and confirmed the concentration-dependent behavior of the B-ECM hydrogels.

Rheological measurements were performed on B-ECM at concentrations of 5, 10 and 15 mg/mL with an ARES-LS2 Shear Rheometer (n = 3). Dynamic time sweep analysis, strain sweep analysis and turbidimetric analysis were performed (S1 Fig). Summary of rheological and turbidimetric measurements shown. Storage modulus is indicated by G', and loss modulus is indicated by G". Lag time of B-ECM hydrogels is defined as the intercept point of the slope at log $t_{1/2}$ and turbidimetry baseline with 0% absorbance. $t_{1/2}$ and $t_{95}$ are defined as the time point when 50% or 95% of maximum absorbance is measured. The gelation rate S is defined as the slope at the linear region of the gelation curve.

## Growth of cerebral organoids on porcine B-ECM and Matrigel

Embryoid body formation and neural induction were performed following the standardized protocol of the "STEMdiff Cerebral Organoid Kit", based on previously published protocols [39]. During the expansion step, EBs were either included in Matrigel or in 15 mg/mL B-ECM hydrogel, a concentration range that showed the best mechanical support. During the following three days, differences in development were observed between these two groups (Fig 5). While Matrigel-surrounded brain organoids show neural budding [1], which gave them a cloud-like shape during early development, the B-ECM hydrogel brain organoids showed a more even growth without budding, but seemed to be slightly more transparent. Although neuronal budding was not observed, ventricular like zones, which usually emerge during neuronal budding, were numerously forming, which indicates correct neuroectodermal development in the B-ECM hydrogel brain organoids (Fig 5B) [2]. At a later stage during maturation (around day 20), when the buds of the Matrigel-grown organoids developed into a smoother morphology, the difference between the groups was less apparent. On day 40, when organoids were fixed and prepared for analysis, no significant morphological difference was visible. RT-qPCR was performed to observe the effect of B-ECM hydrogel or Matrigel on gene expression during brain organoid development. We chose NESTIN, TUBB3, DCX, MAP2 and GFAP as relevant neural markers. No significant differences in gene expression were found, indicating equal effect of Matrigel or B-ECM on gene expression (Fig 5B). To visualize the inner structure and to validate our morphological observations, we analyzed both groups with immunohistochemistry. In line with previous studies [1], radial glia cells (stained with SOX2+, Nestin+, PAX6+) were located in the ventricular like zone of the organoids, from where they have been found to start migrating towards the outside while maturing into neurons (stained with β-tubulin III+, DCX+, MAP2+) (Fig 5C). Taken together, these results indicate that both Matrigel and B-ECM hydrogel supported the maturation process of hESCs into brain organoids.

## Discussion

Current methods to generate cerebral organoids often utilize Matrigel as an external scaffold. Matrigel, however, is a nonspecific hydrogel of mouse tumor origin and does not represent the complexity of the brain protein environment. In the present study, we investigated the application of a hydrogel derived from porcine brain ECM to be used as a scaffold for culturing

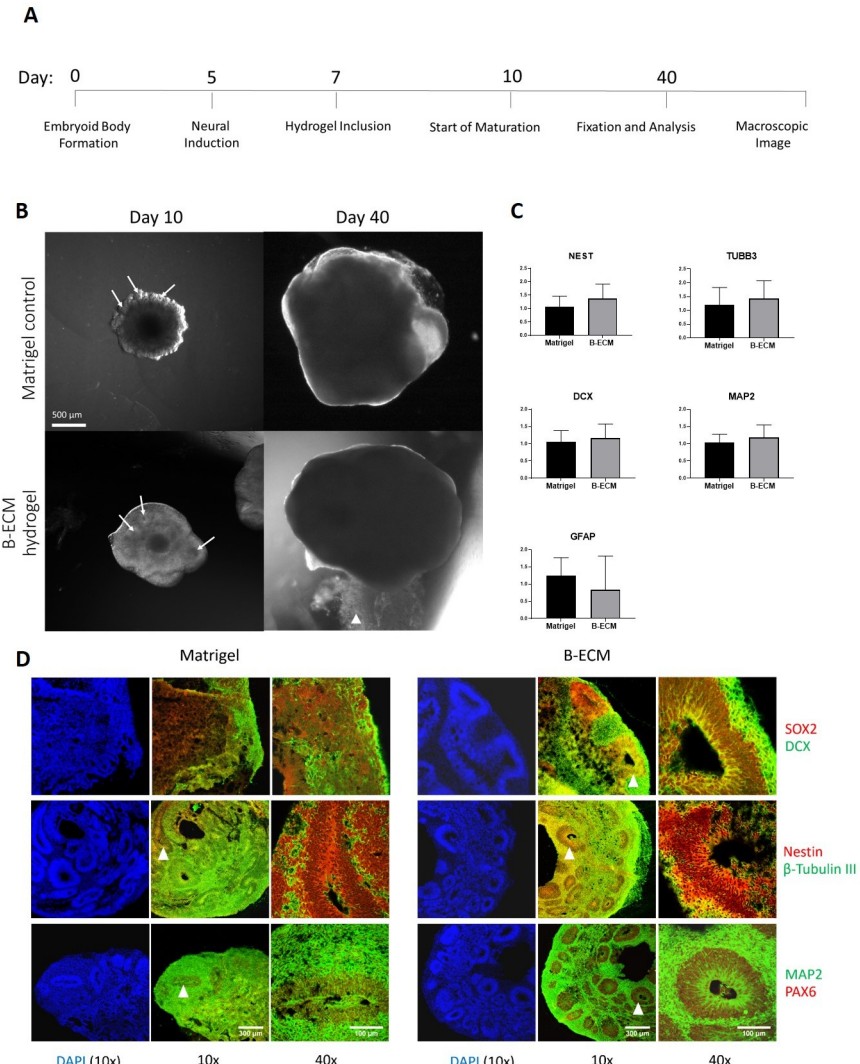

**Fig 5. Growth and gene expression of brain organoids embedded in B-ECM hydrogel or Matrigel.** (A) Timeline of experimental steps of human brain organoid maturation embedded in Matrigel or B-ECM hydrogel. (B) Brain organoids (n = 4) on day 10 in Matrigel (top pictures) or B-ECM hydrogel (bottom pictures). On day 40, no significant difference is apparent. White arrows at day 10 point to ventricular-like zones located in the formed neuronal buds White arrowhead at day 40 indicates residual B-ECM hydrogel. Scale bar: 500 μm. (C) Expression of the genes NEST, TUBB3, DCX, MAP2 and GFAP was measured by extraction of RNA from organoids grown in the presence of B-ECM hydrogel (n = 13) or Matrigel (n = 12) in technical triplicates with RT-qPCR. Statistical analysis performed with unpaired t-test. (D) Tissue sections of brain organoids cultured in Matrigel (left panels) or B-ECM hydrogel (right panels) for 40 days were immunostained for neuronal progenitor markers (SOX2, Nestin, PAX6) and mature neuronal markers (DCX, β-Tubulin III, MAP2). Images were taken with a 10x objective (10x, scale bar: 300 μm) and 40x objective (40x, scale bar: 100 μm). DAPI was used as nuclear counterstaining. White arrowheads indicate ventricular-like zones that were chosen for magnification.

cerebral organoids. We found that the native porcine brain could successfully be decellularized with little or no remaining DNA content, and the prepared hydrogel formed a stable gel starting from concentrations of 5 mg ECM/mL. Functional ECM proteins such as GAGs were partly removed during the DC and hydrogel processing, and the remaining proteins consisted of a high percentage of different collagen types. When B-ECM hydrogels were used as scaffolds for cerebral organoid growth, cerebral organoids were obtained that were similar to the control

group cultured in Matrigel, but with morphologically divergent neuroepithelial development. Taken together, our results show that B-ECM hydrogels are a suitable scaffold for the generation of cerebral organoids and may be further optimized to obtain more specific cerebral organoid development.

Brain decellularization has previously been performed for various applications such as direct injection into the brain [23, 29], in combination with other hydrogel scaffolds [26], disease modeling [28], or as a scaffold for neuronal growth [22]. In the present study, we used brains from pigs for preparation of B-ECM. We chose the pig brain as it is commonly used as a model for human brain in biomedical research, due to morphological, structural and compositional similarities as well as availability [43]. Initially, we tested different DC protocols based on combinations of SDS, TritonX or SDC as well as DNA removing enzymes. From these initial experiments, we developed a new protocol based on the ionic detergent SDC and DNase. SDC proved to be a suitable DC detergent for the large brain volume, with little or no cellular remnants or antigens detectable after DC. The use of SDC-based protocols has previously shown to be beneficial over SDS-based protocols, as increased damage to ECM proteins in different tissue DC applications has been reported with SDS [44–46]. Another advantage of this protocol is that it only applies a single detergent and a single enzyme, which simplifies the evaluation and optimization of the protocol compared to protocols using multiple different detergents for DC [22, 23, 25]. Furthermore, we observed a beneficial effect of cutting the brain into smaller pieces, as larger pieces did not allow for full penetration of the tissue by detergents and required longer incubation times. DC of smaller tissue pieces has previously been shown as beneficial for the removal of lipids in human pancreas [47], which otherwise might interfere with the gelation of the hydrogel. The smaller tissue pieces may be a reason for the observed weight loss of the total brain tissue of 98.7% after DC, due to loss of small tissue particles during filtration. Alternative methods to increase the yield of B-ECM may include smaller filter pores or alternative methods to separate tissue from liquid, such as ultra-centrifugation or lyophilization between liquid changes.

After DC and subsequent acid-pepsin degradation, B-ECM hydrogel formation was initiated after the neutralization of pH and salt concentration, as described previously [31]. While previous publications reported that neutralized pre-hydrogel solution did not gel at 4˚C [37], we observed gelation at this temperature when stored overnight. Gelation was affected by temperature, pH and salt concentration, and already minor adoptions of these parameters can have a strong effect on the hydrogel mechanical properties and gelation kinetics [48]. In some cases, we observed heterogenous or immediate gelation directly after adding NaOH for pH neutralization. To better standardize the hydrogel process, we added a step in which the neutralized hydrogel was frozen down, lyophilized and milled it into a powder. The powder could then be dissolved in water to then form a hydrogel. This method removes time dependence for starting an experiment and allows the formation of a stock of B-ECM material, which can readily be turned into a hydrogel within 30 minutes. Nevertheless, the inconsistency of ECM-based hydrogels is a commonly observed obstacle, which could be optimized with improved hydrogel formation processes [47].

The protein composition of the B-ECM is responsible for its specific function, and the B-ECM contains proteins not found in other tissues of the body. Given the functional implications of the ECM as a stem cell niche, the loss of proteins during the DC process must be monitored. Mass spectrometry results showed that several functional proteins were lost during the hydrogel preparation process, which may affect the growth of organoids in the brain hydrogel. The B-ECM and B-ECM hydrogel group had a high abundance (greater than 90%) of collagens. Collagen is less abundant in brain tissue compared to other tissue types [11], however different collagen types are present in parts of the brain such as the microvasculature [49].

This may explain the remaining collagen content, which has also been reported in other studies performing brain DC [22]. As whole pig brains were used in this study, also collagen from the dura mater and pia mater may be responsible for the observed collagen content [50].

The high abundance of collagens is an issue of common concern in proteomics of decellularized tissues, as it interferes with the detection of low abundant proteins. Therefore depletion of collagens may be considered to facilitate the identification of matrisome-proteins that are masked by the collagens [51]. Nevertheless, the B-ECM retained about 50% of the matrisome-proteins that were observed in the native tissue. Noteworthy, label-free mass spectrometry proteomics has an intractable problem of missing values, which may exceed 50% of the dataset and is a limitation of this method [52]. The sources for the missing data include low abundance, weak ionization, and random sampling, resulting in that peptides that were identified in some samples are not detected in others [53]. Hence, the difference between the native tissue, B-ECM, and B-ECM hydrogel regarding the composition of the matrisome-proteins may be due to a low abundance of proteins and or other technical errors. The remaining ECM proteins observed with IHC were mainly structural proteins such as different types of collagen, laminin, fibronectin, vitronectin. These structural proteins have been shown to mediate and engage in neuronal proliferation and outgrowth [54–56]. Laminin was fragmented in the hydrogel, a result also previously reported [47]. Quantification of collagen showed a remaining content of 113.5 μg/mg dry weight. Pure collagen hydrogels are most commonly applied at a concentration of approximately 2 mg collagen/mL [57], however, gelation can already occur at a concentration of 1 mg/mL or below [58]. This explains the gelation observed in this study, where hydrogels of 15 mg/mL B-ECM contain 1.70 mg/mL of pure collagen. Loss of GAGs in the DC process may in part be explained by the removal of cells, which contain membrane–bound GAGs. Removal of GAGs is seen as unbeneficial, as GAGs can guide cell behavior [59]. However, GAG removal is a common result of detergent/enzyme based DC, and has been observed in multiple different studies [16, 60–62]. Improvement of the DC protocol may lead to a better retention of GAGs in the decellularized tissue [63].

Mechanical properties (e.g. stiffness) are a cellular cue and can guide cell behavior, therefore it is important to analyze the mechanical properties of the hydrogel, which also depend on the decellularization method [64, 65]. Gelation times observed in this study ranged from 20–25 min until gelation was completed. Brain tissue storage moduli, indicating the stiffness of the gel, have been reported between 200 and 1,000 Pa as a target moduli range [41, 66, 67], and neurons have been shown to grow better on hydrogels with a similar stiffness to the native brain [67, 68]. Reported values for storage modulus of decellularized brain ECM typically range between 20 and 60 Pa and loss modulus between 2 and 10 Pa [37]. Similar storage and loss moduli were observed in this study as 15 mg/mL B-ECM hydrogels had a storage modulus of 36 Pa and a loss modulus of 8 Pa. The lower stiffness of the B-ECM hydrogel compared to the native brain tissue may be adjusted by changes in B-ECM concentration or addition of another crosslinking material. However, an increase in concentration, needed to obtain a desired rheological property, may affect the ability of cells to penetrate the hydrogel as well as diffusion of oxygen, nutrients, and waste products across the gel, which may negatively affect the outcome in non-vascularized organoids [69, 70]. We therefore chose a concentration of 15 mg/mL, which showed lower stiffness than the native brain, but provided a suitable environment for organoid growth.

No significant differences in the expression of neurogenic genes between organoids grown in the presence of Matrigel and B-ECM were observed. However, a distinct difference in the morphological development in the early organoid (day 10) was noted. Instead of the characteristic neural-budding of neuroepithelial tissue, B-ECM brain organoids developed more uniformly. The transparent external border is highly enhanced for B-ECM organoids compared

to the Matrigel group. The development of transparent tissue after neural induction is a sign of tissue differentiation into neuroectodermal lineage. Neuroectodermal tissue gives rise to all neuronal cell types and thus beneficial for brain organoid development [1]. Furthermore, the identification of neural progenitor and immature neuronal markers by IHC confirmed a regular maturation process for 40 days old brain organoids in both groups. In our overall assessment, the B-ECM group showed a tendency to have slightly more distinctive ventricular like zones. From these observations, we conclude that the B-ECM produced by this protocol supports the development of neuroepithelial tissue as well as Matrigel and can thus be considered as a suitable alternative hydrogel for developmental studies of the human brain. Further optimization of DC protocols to improve the retention of functional proteins could lead to more enhanced effects of applying B-ECM.

While decellularization is an attractive method to recreate the native cell environment, batch-to-batch variations, variabilities and operator dependency in the process are disadvantages for standardization efforts [71]. Already small alterations in the DC protocol might lead to a different outcome and reduce reproducibility. Also, donor starting material can be variable in terms of animal species, origin and age. Especially the age of the animal could affect neuronal network formation, as the ECM is dynamically remodeled and shows a different protein composition at different life stages [72], as previously shown in a comparison of fetal and adult pig brain ECM [22]. Also in this study, adult pig brains were used, due to good availability, cheap costs and the practical and economical difficulties of utilizing embryonic pig brains. Indeed, embryonic pig brains may be a more suitable model for brain organoid development [22], however the high costs of obtaining embryonic pig brains and ethical concerns may be a practical hurdle for large scale adoption as a model for brain organoid development compared to Matrigel. Nevertheless, this study shows the potential application for decellularized brain hydrogels, and further efforts to reduce variability and optimize DC protocols and hydrogel formation could lead to improved native hydrogels for biomedical applications.

## Conclusions

In this study, we showed that porcine B-ECM hydrogels could be produced from native adult porcine brain, with few host cells remaining. During DC and hydrogel processing, several functional proteins were lost, however, this did not negatively impact organoid utility. After DC, the B-ECM consisted mainly of different collagen types, which is beneficial in terms of hydrogel stability, but may not have the same beneficial functional effect as if more functional, brain-specific proteins were retained. Hence, the process described here can be further optimized to reduce extraction losses and increase the content of functional proteins in the hydrogel. The B-ECM hydrogels were successfully applied as native scaffolds for cerebral organoid formation and were as efficient as Matrigel. Thus, the B-ECM could provide a more native environment, and a more consistent matrix source, for studies that aim to investigate the development of the human brain with organoids.

## Supporting information

**S1 Fig. Rheological characterization of B-ECM hydrogels at different concentrations.** Rheological measurements were performed on B-ECM at concentrations of 5, 10 and 15 mg/mL with an ARES-LS2 Shear Rheometer (n = 3) (A) Dynamic time sweep analysis of hydrogel at different concentrations, showing storage modulus (G') and loss modulus (G") after initiation of gelation. (B) Strain sweep shows the physical failure of the gel at increased strain values between 0.1% to 200%. Gelation kinetics were furthermore observed by turbidimetric analysis. (C) Result of turbidimetric analysis for B-ECM hydrogels at concentrations of 5, 7.5 and 10

mg/mL (n = 6) at 37˚C, measured at 405 nm in a photo spectrometer.
(TIF)

**S2 Fig. Venn-diagramm of overlapping ECM proteins, detected by mass spectrometry.** A Vonn-diagramm, showing the overlapping proteins per group which were detected with mass spectrometry.
(TIF)

**S1 File. Raw data of ECM proteins, analyzed by mass spectrometry.** Excel file of ECM proteins analyzed with mass spectrometry for Matrigel, Native, B-ECM and B-ECM hydrogel groups.
(XLSX)

**S2 File. The normalized and filtered proteomics data, obtained by mass spectrometry.** Excel file of ECM proteins analyzed with mass spectrometry for Matrigel, Native, B-ECM and B-ECM hydrogel groups.
(XLSX)

**S3 File. The normalized and filtered proteomics data, showing the overlapping protein identification.** Excel file of ECM proteins analyzed with mass spectrometry for Matrigel, Native, B-ECM and B-ECM hydrogel groups, corresponding to the Venn-Diagram (S2 Fig).
(XLSX)

**S4 File. The normalized and filtered proteomics data, filtered for matrisome proteins.** Excel file of ECM proteins analyzed with mass spectrometry for Matrigel, Native, B-ECM and B-ECM hydrogel groups.
(XLSX)

**S1 Table. Primers used for the RT-qPCR experiment.** Fw = forward, Rv = reverse.
(DOCX)

## Acknowledgments

We would like to thank Luke Perreault for supporting us with ECM processing and hydrogel formation. We also thank Erwin Schoof for his critical reading of the manuscript and for providing access to mass spectrometry facilities. The qPCR experimental development and data analysis were provided by the Genomics and NGS Core Facility at the Centro de Biología Molecular Severo Ochoa (CBMSO, CSIC-UAM) which is part of the CEI UAM+CSIC, Madrid, Spain.

## Author Contributions

**Conceptualization:** Robin Simsa, David L. Kaplan, Per Fogelstrand.

**Data curation:** Robin Simsa, Theresa Rothenbücher, Hakan Gürbüz, Nidal Ghosheh.

**Formal analysis:** Robin Simsa, Theresa Rothenbücher, Nidal Ghosheh.

**Investigation:** Robin Simsa, Theresa Rothenbücher, Hakan Gürbüz, Nidal Ghosheh.

**Methodology:** Robin Simsa, Theresa Rothenbücher, Hakan Gürbüz, Nidal Ghosheh, Per Fogelstrand.

**Project administration:** Robin Simsa, Lachmi Jenndahl, David L. Kaplan, Alberto Martinez Serrano, Per Fogelstrand.

**Resources:** Jenny Emneus, Lachmi Jenndahl, David L. Kaplan, Per Fogelstrand.

**Supervision:** Jenny Emneus, Lachmi Jenndahl, David L. Kaplan, Niklas Bergh, Alberto Martinez Serrano, Per Fogelstrand.

**Validation:** Robin Simsa, Theresa Rothenbücher, Nidal Ghosheh.

**Visualization:** Robin Simsa, Theresa Rothenbücher.

**Writing – original draft:** Robin Simsa, Theresa Rothenbücher, Hakan Gürbüz, Nidal Ghosheh, Per Fogelstrand.

**Writing – review & editing:** Robin Simsa, Theresa Rothenbücher, Jenny Emneus, Lachmi Jenndahl, David L. Kaplan, Niklas Bergh, Alberto Martinez Serrano, Per Fogelstrand.

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
