## [Decision Letter · Decision Letter 0]

9 Oct 2020

PONE-D-20-23633

Brain organoid formation on decellularized porcine brain ECM hydrogels

PLOS ONE

Dear Dr. Simsa,

Thank you for submitting your manuscript to PLOS ONE. After careful consideration, we feel that it has merit but does not fully meet PLOS ONE’s publication criteria as it currently stands. Therefore, we invite you to submit a revised version of the manuscript that addresses the points raised during the review process.

We look forward to receiving your revised manuscript.

Kind regards,

Nic D. Leipzig

Academic Editor

PLOS ONE

Journal Requirements:

2. We understand that you purchased animal tissue from Amazon.com for this study. In your Methods section, please provide additional regarding the source of this material, specifically any details about the purchased items (e.g., lot number, source origin, description of appearance) that will facilitate reproducibility of the analyses.

"This project has received funding from the European Union’s Horizon 2020 research and innovation

594 program under the Marie Skłodowska-Curie grant agreement No 722779 and was conducted within the

595 “Training 4 Cell Regenerative Medicine” (T4CRM) network. Work at AMS group (CBMSO-UAM) was also

596 supported by grants MINECO SAF-2017-83241-R, and ISC-III RETICS TerCel RD16/0011/0032 and ASCTN597

Training (No. 813851). Work at Tufts was supported by the NIH (P41EB027062 and R01NS092847)".

i) We note that you have provided funding information that is not currently declared in your Funding Statement. However, funding information should not appear in the Acknowledgments section or other areas of your manuscript. We will only publish funding information present in the Funding Statement section of the online submission form.

ii) Please remove any funding-related text from the manuscript and let us know how you would like to update your Funding Statement. Currently, your Funding Statement reads as follows:

Additional Editor Comments (if provided):

Also, to add to the reviewers comments, my biggest question with this work and the materials used is do the authors include the dura and the pia in the harvested brains and does that ECM carry over the matrix that is created form them? The reason I mention this is that the parenchyma of the CNS is well documented to contain very little matrix that is almost devoid of fibrillary matrix (especially fibrillary collagens), and it is mostly cells by volume (nearly 75% by some reports) (e.g., Zimmermann, 2008, DOI: 10.1007/s00418-008-0485-9). Thus the B-ECM hydrogels used are not necessarily what the authors think they are, and definitely not what they are being represented as. To me they are a mixed up version of the adult pig native brain and its associated basement membranes – I would expect nearly none of the collagens that Fig 3 proteomic analysis shows if purer white/grey matter was used. To be honest using a purer source would probably be hard to gel natively. So the bigger question in terms of the mixed up ECM nature of the B-ECM hydrogels is this a good or a bad thing in terms of desired application for brain organoids?

My other point I would like to add is the data shown (beyond Fig.1) appears to be relatively devoid of statistical analyses.

Please fully address all reviewer and my comments if you wish to resubmit.

Reviewers' comments:

Reviewer's Responses to Questions

**Comments to the Author**

1. Is the manuscript technically sound, and do the data support the conclusions?

Reviewer #1: Yes

Reviewer #2: Partly

2. Has the statistical analysis been performed appropriately and rigorously? 

Reviewer #1: Yes

Reviewer #2: Yes

3. Have the authors made all data underlying the findings in their manuscript fully available?

Reviewer #1: Yes

Reviewer #2: Yes

4. Is the manuscript presented in an intelligible fashion and written in standard English?

Reviewer #1: Yes

Reviewer #2: Yes

5. Review Comments to the Author

Reviewer #1: This is an interesting and well written paper.

It would benefit from some key clarifications as there are multiple ECM endpoints examined that need better grouping, rationale and detail:

-Pig brains were decellularized, but appear to retain intravascular contents (ie they were not perfused) correct? This is a key point.

-Fig 1 collagen content should define which collagens are detected with the kit and whether that is relevant to brain.

-Similarly, GAGs measured with the kit used are only sulfated GAGs, this should be noted. Also was HA (nonsulfated GAG) measured? why not?

-Fig 2 the choices for IHC are not well justified. The authors should explain why those ECM molecules were chosen. If focusing on basement membrane, then why BMP and HLA-DR? Vitronectin seems out of place here.

-Fig 3 is matrisome analysis, but should discuss the molecules based on their relevance and location in the brain.

-In addition to noting whether the ECM molecules of interest are relevant to vasculature (whether glycocalyx or basement membrane) or brain parenchyma ECM, a summary of the ECM findings in chart form might be helpful.

Reviewer #2: In this paper, Simsa and colleagues have introduced a decellularized porcine brain extracellular matrix (B-ECM) as an advanced alternative scaffold for human embryonic stem cell (hESC)-derived brain organoids. Compared to matrigel, which is conventionally used to provide structural support and biological cues for the brain organoid maturation, B-ECM is known to have more similar and abundant components of brain extracellular matrix. The authors first analyzed the properties of B-ECM hydrogels, including their composition, surface structure, and viscoelastic behavior, and compare with those of matrigel, showing that B-ECM is suitable as a biocompatible scaffold for brain organoid culture. Next, they embedded the cerebral organoids in B-ECM hydrogel and observed smoother neural epithelial development than those in matrigel, suggesting that B-ECM indeed provides substantial structural support for the growth of brain organoids. Although the authors well-presented analytical data on B-ECM hydrogel for brain organoid culture, their data is insufficient to argue that B-ECM provides more advantageous growth environment than matrigel. The results merely proved that the usage of B-ECM can yield the similar level of maturation process of brain organoids compared to matrigel.

Major comments

(1) If there was no significant differences observed during the maturation stage of organoid when either B-ECM or matrigel were used, why should researchers use B-ECM hydrogel rather than matrigel? Also, why does B-ECM induce smoother neural epithelial development compared to matrigel? These questions should be addressed in order to persuade the readers that B-ECM is more beneficial to use for brain organoid culture.

(2) The components of B-ECM is rather different than what authors initially mentioned. B-ECM is known to have more glycosaminoglycans (GAGs) and includes less portion of fibrous proteins, such as collagen. However, their analytical data showed that their B-ECM is highly enriched with collagen while the percentage of proteoglycans and glycoproteins is very low compared to matrigel. Indeed, in Figure 3A, matrigel has more ‘ideal’ composition rather than B-ECM hydrogel. Why is the author’s B-ECM possess such a low level of GAGs components?

(3) They introduced the ‘novel’ methodology to synthesize a decellularized B-ECM. How is the novel method more advantageous over the previous methods? More details on the methodologies and comparison with others should be addressed in order to persuade the readers that the methods described here perform well.

Minor comments

(1) - There is no information about the pig’s brain described such as age, sex, the region of the brain, and presence of inflammation. There might be some difference in ECM composition between young and old pig brains.

(2) The spelling or typos must be checked. For example, in line 546, ‘therefor’ should be changed to ‘therefore.’

(3) In figure 4, there are no letter labels (A, B, C). Please check the page 10, line 365 and 369.

(4) In page 14, “DNAse” -> “DNase”

6. PLOS authors have the option to publish the peer review history of their article (what does this mean?). If published, this will include your full peer review and any attached files.

Reviewer #1: No

Reviewer #2: No

---

## [Author Response · Author response to Decision Letter 0]

15 Dec 2020

Dear Editor and Reviewers,

Thank you for your comments, they were addressed in the attached file "Response to Reviewers".

Thank you and best regards,

Robin Simsa

---

## [Decision Letter · Decision Letter 1]

24 Dec 2020

PONE-D-20-23633R1

Brain organoid formation on decellularized porcine brain ECM hydrogels

PLOS ONE

Dear Dr. Simsa,

Thank you for submitting your manuscript to PLOS ONE. After careful consideration, we feel that it has merit but does not fully meet PLOS ONE’s publication criteria as it currently stands. Therefore, we invite you to submit a revised version of the manuscript that addresses the points raised during the review process.

Thank  you for the revision, overall the manuscript is much improved.  Reviewer 1 brings up some important points I agree with - namely making sure that the abstract and the conclusion convey the importance of collagen in the B-ECM treatments in this study. I will add an additional concern about this as well, namely that an adult ECM is being used in an attempt to drive an early embryonic process. We all hate Matrigel, but its tumorigenic-derived composition may actually be closer to an early embryonic matrix composition then any decellularized (DC) adult tissue. The last paragraph of the discussion kind of points this out, but I think itneeds to be more explicit in terms of brain organoids and their significant developmental timeline differences from the DC tissues used.

We look forward to receiving your revised manuscript.

Kind regards,

Nic D. Leipzig

Academic Editor

PLOS ONE

Additional Editor Comments (if provided):

Thank you for the revision, overall the manuscript is much improved. Reviewer 1 brings up some important points I agree with - namely making sure that the abstract and the conclusion convey the importance of collagen in the B-ECM treatments in this study. I will add an additional concern about this as well, namely that an adult ECM is being used in an attempt to drive an early embryonic process. We all hate Matrigel, but its tumorigenic-derived composition may actually be closer to an early embryonic matrix composition then any decellularized adult tissue. The last paragraph of the discussion kind of points this out, but I think needs to be more explicit in terms of brain organoids and their significant developmental timeline differences from the DC tissues used.

Reviewers' comments:

Reviewer's Responses to Questions

**Comments to the Author**

1. If the authors have adequately addressed your comments raised in a previous round of review and you feel that this manuscript is now acceptable for publication, you may indicate that here to bypass the “Comments to the Author” section, enter your conflict of interest statement in the “Confidential to Editor” section, and submit your "Accept" recommendation.

Reviewer #1: (No Response)

Reviewer #2: All comments have been addressed

2. Is the manuscript technically sound, and do the data support the conclusions?

Reviewer #1: Yes

Reviewer #2: Yes

3. Has the statistical analysis been performed appropriately and rigorously? 

Reviewer #1: Yes

Reviewer #2: Yes

4. Have the authors made all data underlying the findings in their manuscript fully available?

Reviewer #1: Yes

Reviewer #2: Yes

5. Is the manuscript presented in an intelligible fashion and written in standard English?

Reviewer #1: Yes

Reviewer #2: Yes

6. Review Comments to the Author

Reviewer #1: The authors did a commendable job of addressing concerns. However, given the dominance of collagen in the final product, the manuscript needs to be revised to reflect this. At present there are just side references to this collagen-dominant composition in the results and discussion with little attention to the implications for utilization of the organoid model. The abstract and conclusion do not even mention the collagens. A different perspective in the writing is warranted. There are also a few grammatical errors like the use of the word microvascular, which is an adjective, as a noun.

Reviewer #2: All of my previous concerns were well addressed. The revised manuscript is well deserved to be published in PLoS One.

7. PLOS authors have the option to publish the peer review history of their article (what does this mean?). If published, this will include your full peer review and any attached files.

Reviewer #1: No

Reviewer #2: **Yes: **Ki-Jun Yoon

---

## [Author Response · Author response to Decision Letter 1]

25 Dec 2020

Thank you for your thorough comments. We have adopted our manuscript accordingly and answered to them directly in the "Response to Reviewers" document.

Best regards,

Dr. Robin Simsa

---

## [Editor Report · Decision Letter 2]

6 Jan 2021

Brain organoid formation on decellularized porcine brain ECM hydrogels

PONE-D-20-23633R2

Dear Dr. Simsa,

We’re pleased to inform you that your manuscript has been judged scientifically suitable for publication and will be formally accepted for publication once it meets all outstanding technical requirements.

Kind regards,

Nic D. Leipzig

Academic Editor

PLOS ONE
---

## [Editor Report · Acceptance letter]

11 Jan 2021

PONE-D-20-23633R2 

Brain organoid formation on decellularized porcine brain ECM hydrogels 

Dear Dr. Simsa:

I'm pleased to inform you that your manuscript has been deemed suitable for publication in PLOS ONE. Congratulations! Your manuscript is now with our production department. 

Kind regards, 

on behalf of

Dr. Nic D. Leipzig 

Academic Editor

PLOS ONE